# The mediating role of insomnia symptoms in the association between perceived neighborhood danger and depressive symptoms in later life

Seo-Yun Choi[1], Yuri Jang[1,2]*

1 Edward R. Roybal Institute on Aging, Suzanne Dworak-Peck School of Social Work, University of Southern California, Los Angeles, California, United States of America, 2 Department of Social Welfare, Ewha Womans University, Seoul, Republic of Korea

* yurij@usc.edu

## Abstract

Perceived neighborhood danger poses significant risks to mental health in later life, yet the underlying psychological mechanisms remain unclear. Given that environmental stressors can compromise sleep health, we conceptualize insomnia symptoms as a potential mediator in the association between perceived neighborhood danger and depressive symptoms. Data from 2,240 adults aged 65 or older from Wave 3 of the National Social Life, Health, and Aging Project (NSHAP) were analyzed. Multivariate analyses showed significant associations of both perceived neighborhood danger and insomnia symptoms with depressive symptoms after controlling for sociodemographic and health covariates. Bootstrap mediation analysis provided supportive evidence for the indirect effect of perceived neighborhood danger on depressive symptoms through insomnia symptoms (B = .04, SE = .01, bias-corrected 95% CI = [.02, .07]), accounting for 25.8% of the total effect. Insomnia symptoms were identified as an important pathway through which perceived neighborhood danger undermines mental health in older adults.

## Introduction

Research on health determinants has increasingly recognized the importance of factors that extend beyond the individual level. The National Institute on Minority Health and Health Disparities (NIMHD) Research Framework [1] calls for greater attention to community-level determinants of health, and the model proposed by Diez Roux and Mair [2] illustrates how neighborhood social environments influence health through behavioral mechanisms and stress-related processes. In line with these frameworks, the present study focuses on perceived neighborhood danger as a key neighborhood environmental factor and addresses its health impact in later life.

Defined as residents' subjective perceptions of danger in their local environment [3], perceived neighborhood danger is particularly relevant to sleep and mental health

**Data availability statement:** All data files are available from the NSHAP database. https://www.icpsr.umich.edu/sites/view/studies/34921/explore-data.

**Funding:** The author(s) received no specific funding for this work.

**Competing interests:** The authors have declared that no competing interests exist.

because it reflects individuals' lived experiences and cognitive appraisals of threat. When individuals perceive their surroundings as unsafe or dangerous, they often experience heightened vigilance and anxiety, which can undermine sleep quality and overall well-being. It is noteworthy that subjective perceptions of the neighborhood environment exert stronger influences on mental health than objective indicators such as crime statistics (e.g., crime and arrest rates) and physical disorder (e.g., graffiti, vandalism, abandoned buildings) [4–6]. Furthermore, older adults are known to be more susceptible to neighborhood influences than their younger counterparts, as they spend more time within their neighborhoods [3]. However, the pathways linking perceived neighborhood danger to depressive symptoms in later life remain largely unexplored.

Filling this gap, we conceptualize sleep problems as an intermediate factor in the association between perceived neighborhood danger and depressive symptoms. Among various types of sleep problems, we focus on insomnia symptoms given their high prevalence and relevance to the current investigation. Referring to difficulties initiating or maintaining sleep, or experiencing poor-quality sleep [7], insomnia symptoms are frequently experienced by individuals who undergo age-related changes in sleep and circadian rhythms [8]. The particular link between insomnia symptoms and depression has been widely documented across diverse groups of older adults [9,10]. Neighborhood research also suggests that worries and concerns related to neighborhood danger make older individuals prone to insomnia symptoms [11,12].

Taken together, it is plausible to anticipate that the level of neighborhood danger perceived by older individuals is associated with insomnia symptoms, which in turn lead to depressive symptoms. This mediating conceptualization aligns with stress literature and pathway models linking neighborhood environments to mental health outcomes [13–17]. In the present study, using a national sample of older adults, we aim to examine the direct effect of perceived neighborhood danger and insomnia symptoms on depressive symptoms and the mediating role of insomnia symptoms in the association between perceived neighborhood danger and depressive symptoms. Findings from this investigation would help improve understanding of how neighborhood environments shape older adults' sleep and mental health, as well as the psychological mechanisms underlying these relationships, and may offer implications for health promotion strategies.

## Methods

### Sample

This study utilized data from Wave 3 of the National Social Life, Health and Aging Project (NSHAP), conducted in 2015–2016, which includes a nationally representative sample of community-dwelling adults in the United States [18]. The original data collection was IRB-approved with participant informed consent, and secondary analysis was exempt from additional IRB review. The sample for the current investigation was selected through step-by-step exclusions from the original 4,377 participants. First, we excluded participants younger than 65 years, leaving 2,539 participants. Next, we excluded 236 participants who did not complete the leave-behind

questionnaire which includes questions on neighborhood and sleep health. Finally, 63 participants were excluded due to missing responses for race (n = 8) or selection of the "other" race category (n = 55). The final sample included 2,240 older adults.

## Measures

***Perceived neighborhood danger.*** Perceived neighborhood danger was measured with three statements ("people in this area are afraid at night," "there are places where 'trouble' is expected," and "you are taking a big chance walking alone at night"), selected from the Project on Human Development in Chicago Neighborhoods-Community Survey (PHDCN-CS) and validated in previous studies [3,19–21]. Respondents were asked to indicate their level of agreement on each statement using a 5-point scale (1 = strongly disagree to 5 = strongly agree). Total scores ranged from 3 to 15, with higher scores indicating a greater level of perceived neighborhood danger. The internal consistency of the scale in the present sample was high ($\alpha$ = .83).

***Insomnia symptoms***. Insomnia symptoms were assessed using three items ("trouble falling asleep," "trouble waking up during the night," and "trouble waking up too early and not being able to fall asleep again"), which correspond to the cardinal clinical features of insomnia defined in the Diagnostic and Statistical Manual of Mental Disorders, Fifth Edition (DSM-5) [22]. These items have been used to assess insomnia symptoms in prior NSHAP research [23]. Respondents reported the frequency of experiencing each symptom on a 4-point scale (0 = never to 3 = most of the time). Total scores ranged from 0 to 9, with higher scores indicating more frequent insomnia symptoms. The scale demonstrated marginally acceptable internal consistency ($\alpha$ = .68).

***Depressive symptoms***. Depressive symptoms were assessed using the NSHAP Depressive Symptoms Measure (NDSM) [24], an 11-item short form adapted from the Center for Epidemiologic Studies Depression Scale (CES-D). Items were rated on a 3-point scale (0 = rarely or none of the time to 2 = occasionally or most of the time). Example items include feelings of sadness, hopelessness, loneliness, lack of motivation, difficulty concentrating, and feeling like a failure. Total scores range from 0 to 21, higher scores indicating greater levels of depressive symptoms. Internal consistency of the scale was high ($\alpha$ = .79).

***Covariates.*** Based on previous research on neighborhood environment, sleep, and mental health in the older adult population [11,19,25,26], sociodemographic and health variables were selected as covariates. Sociodemographic characteristics included age (in years), gender (0 = male, 1 = female), race (0 = Non-Hispanic White, 1 = Non-Hispanic Black, 2 = Hispanic), education (0 = High school graduation or below, 1 = some college or above), marital status (0 = not married/partnered, 1 = married/partnered), subjective financial status (0 = below average relative to American families, 1 = average or above), and residential duration (1 = less than one year to 5 = more than 50 years).

Health-related variables included chronic condition and functional limitation. Chronic condition was measured with a checklist including hypertension, arthritis, heart condition, cancer, diabetes, respiratory conditions, and stroke. Functional limitations were measured by six items of the Activities of Daily Living (walking across a room, dressing, bathing, eating, getting in/out of bed, using the toilet) and another six items of the Instrumental Activities of Daily Living (preparing meals, taking medications, managing money, shopping, housework, using the telephone). Internal consistency of the scale was high ($\alpha$ = .89), and total scores ranged from 0 to 12, with higher scores indicating greater levels of functional limitation.

## Analytic strategy

After reviewing the descriptive statistics, we examined bivariate correlations among study variables to understand the underlying associations among the study variables and ensure the absence of collinearity. We then conducted multivariate linear regression analysis to test the direct effects of perceived neighborhood danger and insomnia symptoms on depressive symptoms. In the initial model, the effect of perceived neighborhood danger was tested after controlling for the set of covariates (age, gender, race/ethnicity, education, marital status, subjective financial status, resident duration, chronic

conditions, and functional limitations). Subsequently, insomnia symptoms were added to the model, which was intended to examine its direct effect, as well as potential role as a mediator. Ultimately, the indirect effect model was tested using PROCESS macro [27]. The 95% confidence interval was estimated using 5,000 bootstrap resamples. Analyses were performed using SPSS Statistics 29 (IBM Corp., Armonk, NY).

## Results

### Descriptive characteristics of the sample

Table 1 summarizes sample characteristics. The mean age of the participants was 75.3 years (SD = 7.03), and more than half were female. About 65.3% of the participants were married/partnered and 60.1% had some college education or above. The sample was predominantly Non-Hispanic White (76.4%), followed by Non-Hispanic Black (13.5%) and Hispanic (9.96%). About 72.7% reported average or above financial status. About 57.3% of participants had lived in their neighborhood for more than 20 years. On average, the sample had 1.55 chronic medical conditions (SD = 1.25) and 1.81 functional limitations (SD = 3.99). The mean scores for perceived neighborhood danger, insomnia symptoms, and depressive symptoms, were 7.06 (SD = 2.68), 4.80 (SD = 2.04), and 4.79 (SD = 4.29), respectively.

**Table 1. Descriptive Characteristics of Study Sample.**

|  | M ± SD, % | Minimum-maximum |
|---|---|---|
| Age | 75.3 ± 7.03 | 65-95 |
| Gender |  |  |
| Male | 44.9 |  |
| Female | 55.1 |  |
| Race/Ethnicity |  |  |
| White | 76.4 |  |
| Black | 13.5 |  |
| Hispanic | 9.96 |  |
| Education |  |  |
| High school graduation or below | 39.9 |  |
| Some college or above | 60.1 |  |
| Marital Status |  |  |
| Married/partnered | 65.3 |  |
| Not married/partnered | 34.7 |  |
| Subjective financial status |  |  |
| Below average | 27.3 |  |
| Average or above | 72.7 |  |
| Residential duration |  |  |
| 20 years or less | 42.7 |  |
| More than 20 years | 57.3 |  |
| Chronic conditions | 1.55 ± 1.25 | 0-7 |
| Functional limitations | 1.81 ± 3.99 | 0-34 |
| Perceived neighborhood danger | 7.06 ± 2.68 | 3-15 |
| Insomnia symptoms | 4.80 ± 2.04 | 0-9 |
| Depressive symptoms | 4.79 ± 4.29 | 0-21 |

## Bivariate correlations of study variables

Table 2 presents the bivariate correlations among study variables. Perceived neighborhood danger was positively correlated with insomnia symptoms ($r = .08$, $p < .01$) and depressive symptoms ($r = .15$, $p < .001$). The strongest association was between depressive symptoms and insomnia symptoms ($r = .32$, $p < .001$). Among the covariates, functional limitations showed positive correlations with all three primary variables ($r = .09$ to $.28$, all $p < .01$). Lower levels of education ($r = -.26$, $p < .001$) and income ($r = -.20$, $p < .001$) were significantly associated with perceptions of neighborhood danger.

## Multivariate regression models of depressive symptoms

Table 3 presents the regression models of depressive symptoms. Variance inflation factors (VIF) for all predictors ranged from 1.03 to 1.22, indicating no multicollinearity concerns. In Model 1, the effect of perceived neighborhood danger on depressive symptoms was significant ($B = .17$, $SE = .04$, $p < .001$) after controlling for all covariates. In Model 2, with the addition of insomnia symptoms, the coefficient for perceived neighborhood danger decreased but remained significant ($B = .13$, $SE = .04$, $p < .01$). This attenuation suggests that insomnia symptoms may serve as a potential mediator linking perceived neighborhood danger to depressive symptoms.

## The mediation effect of insomnia symptoms

The indirect effect of insomnia symptoms was further explored using PROCESS macro. As shown in Fig 1, the indirect effect of perceived neighborhood danger on depressive symptoms through insomnia symptoms was significant ($B = .04$, $SE = .01$, bias-corrected 95% CI = [.02, .07]). The indirect effect accounted for 25.8% of the total effect, indicating a modest but meaningful pathway through insomnia symptoms.

**Table 2. Bivariate Correlations among Study Variables.**

| | 1 | 2 | 3 | 4 | 5 | 6 | 7 | 8 | 9 | 10 | 11 | 12 | 13 |
|---|---|---|---|---|---|---|---|---|---|---|---|---|---|
| 1. Age | – | | | | | | | | | | | | |
| 2. Gender (Male) | .02 | – | | | | | | | | | | | |
| 3. Race (vs. White) - Black | .02 | .06** | – | | | | | | | | | | |
| 4. Race (vs. White) - Hispanic | −.04* | −.03 | −.12*** | – | | | | | | | | | |
| 5. Education (≤ High school) | −.13*** | −.04 | −.09*** | −.16*** | – | | | | | | | | |
| 6. Marital Status (Not married) | −.23*** | −.28*** | −.14*** | .01 | .08** | – | | | | | | | |
| 7. Subjective financial status (<Average) | .01 | −.07** | −.03 | −.14*** | .21*** | .22*** | – | | | | | | |
| 8. Residential duration (>20 years) | .09*** | .03 | .01 | −.03 | −.07** | .04* | .08*** | – | | | | | |
| 9. Chronic conditions | .06** | −.04 | .07*** | .01 | −.10*** | −.07** | −.11*** | −.03 | – | | | | |
| 10. Functional limitations | .25*** | .04* | .07** | .03 | −.11*** | −.13*** | −.11*** | −.04 | .26*** | – | | | |
| 11. Perceived neighborhood danger | .14*** | .09*** | .20*** | .18*** | −.26*** | −.14*** | −.20*** | .03 | .13*** | .14*** | – | | |
| 12. Insomnia symptoms | .04 | .12*** | −.05* | −.08*** | .01 | −.03 | −.04 | −.01 | .10*** | .09*** | .08** | – | |
| 13. Depressive symptoms | .10*** | .13*** | −.01 | .01 | −.06** | −.16*** | −.15*** | −.02 | .22*** | .28*** | .15*** | .32*** | – |

*$p < .05$. ** $p < .01$. *** $p < .001$.

**Table 3. Direct Effect Models of Depressive Symptoms.**

| | Model 1 | | Model 2 | |
|---|---|---|---|---|
| | B(SE) | t | B(SE) | t |
| Perceived neighborhood danger | .17(.04) | 4.01*** | .13(.04) | 3.10** |
| Insomnia symptoms | | | .65(.05) | 12.89*** |
| Age | .02(.02) | 1.06 | .01(.02) | 0.89 |
| Gender (Male) | .81(.22) | 3.73*** | .49(.21) | 2.33* |
| Race (White) | | | | |
| Black | −1.2(.33) | −3.63*** | −.82(.31) | −2.62** |
| Hispanic | −.46(.37) | −1.25 | .00(.35) | 0.01 |
| Education (≤ High school) | .18(.23) | 0.80 | .12(.22) | 0.56 |
| Marital Status (Not married) | −.66(.24) | −2.74** | −.76(.23) | −3.29** |
| Subjective financial status (<Average) | −.87(.25) | −3.48*** | −.81(.24) | −3.35*** |
| Residential duration | −.04(.05) | −0.85 | −.03(.05) | −0.60 |
| Chronic conditions | .62(.09) | 7.11*** | .52(.08) | 6.11*** |
| Functional limitations | .29(.03) | 10.00*** | .26(.03) | 9.56*** |
| R² | .15 | | .22 | |
| F | 30.67*** | | 44.44*** | |

* $p < .05$. ** $p < .01$. *** $p < .001$.

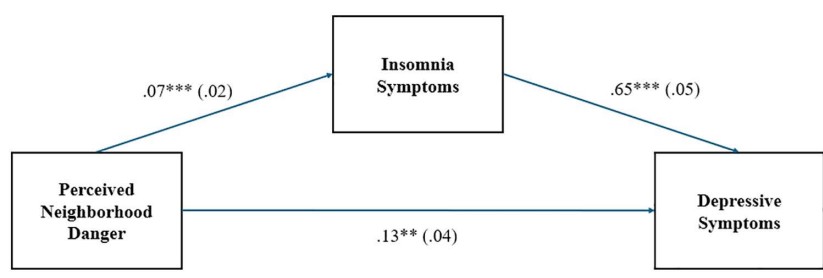

Indirect effect: B = .04, SE = .01, 95% CI = [.02, .07]
*Note*: Age, gender, race, education, marital status, subjective financial status, chronic conditions, functional limitations, and residential duration were controlled for the analysis. ** $p < .01$. *** $p < .001$.

**Fig 1. Mediation model linking perceived neighborhood danger, insomnia symptoms, and depressive symptoms.**

## Discussion

The environments in which we live influence our health and quality of life in ways that extend well beyond what is outwardly visible. For older adults, the surrounding environment can foster a sense of security and belonging or, conversely, pose a risk to mental health [3]. In neighborhoods perceived as dangerous, sleep problems such as insomnia symptoms may arise as a response to heightened vigilance and stress, which can increase susceptibility to depressive symptoms. Guided by theoretical frameworks from environmental health and the stress process model [1,2,13–17], we hypothesized that the mental health impact of perceived neighborhood danger would be mediated by increased insomnia symptoms. Our findings provide supportive evidence for this pathway.

Descriptive analyses showed moderate levels of perceived neighborhood danger and insomnia symptoms, while depressive symptoms exhibited greater variability among participants. Bivariate correlations revealed positive associations

among the three major study constructs, with correlation magnitudes suggesting that collinearity was not a concern. Multivariate analyses further showed that both perceived neighborhood danger and insomnia symptoms were associated with higher levels of depressive symptoms, supporting prior evidence of their mental health risks [6,9]. Overall, these findings underscore the interconnected nature of neighborhood context, sleep health, and depressive symptoms.

This study uniquely examined insomnia symptoms as a mediator, revealing a significant indirect effect of perceived neighborhood danger on depressive symptoms through sleep problems. This finding highlights a potential psychological mechanism in which concerns about neighborhood danger increase vigilance and arousal, impairing sleep initiation and maintenance. Compromised sleep may subsequently weaken emotion regulation and amplify negative cognitive biases, contributing to depressive symptoms. In addition, older adults may limit outdoor activities in neighborhoods perceived as unsafe, reducing exposure to natural light, physical activity, and social interaction, all of which can further disrupt sleep-wake cycles and exacerbate depressive symptoms. Accounting for 25.8% of the total effect, the indirect effect was significant but modest, suggesting that insomnia symptoms represent one pathway linking perceived neighborhood danger to depressive symptoms, while other mechanisms warrant further investigation.

Several limitations should be considered when interpreting these findings. First, the cross-sectional design limits causal inference, and reverse or reciprocal relationships are possible; longitudinal studies are needed to clarify directionality. Second, one depressive symptom item pertains to sleep. To address this overlap, we conducted a sensitivity analysis excluding the sleep-related item in the depressive symptom measure ("My sleep was restless"). The results remained the same, supporting the robustness of the findings independent of measurement overlap. Third, insomnia symptoms were measured via self-report without objective assessment. Future research should incorporate objective measures, such as actigraphy, to complement self-report data. Finally, future work should investigate broader neighborhood and sleep-related factors, as well as alternative pathways that may influence mental health outcomes.

Despite these limitations, the present findings offer several directions for future research and practice. The results suggest that both neighborhood environments and sleep health should be considered when addressing the mental health of older adults. Community-level interventions that enhance both actual and perceived neighborhood safety may serve as a critical first step in supporting mental well-being of older residents. Complementary sleep-focused interventions, such as cognitive behavioral therapy for insomnia (CBT-I) or community-based sleep education, may further support the mental health of those living in adverse neighborhood environments. Future research should explore ways to systematically assess and promote sleep health among older adults in high-risk environments. Additionally, emerging evidence suggests that positive neighborhood characteristics, such as neighborly support and community social cohesion, may protect mental health through pathways involving sleep [26]. Overall, these findings underscore the importance of considering perceived neighborhood risks and insomnia symptoms together when examining mental health in later life.

## Acknowledgments

None.

## Author contributions

**Conceptualization:** Seo-Yun Choi, Yuri Jang.

**Data curation:** Seo-Yun Choi.

**Formal analysis:** Seo-Yun Choi.

**Investigation:** Seo-Yun Choi.

**Methodology:** Seo-Yun Choi.

**Supervision:** Yuri Jang.

**Validation:** Yuri Jang.

**Writing – original draft:** Seo-Yun Choi.

**Writing – review & editing:** Seo-Yun Choi, Yuri Jang.

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
