## [Decision Letter · Decision Letter 0]

10 Feb 2026

PONE-D-25-66548The mediating role of sleep disturbance in the association between perceived neighborhood danger and depressive symptoms in later lifePLOS One

Dear Dr. Jang,

Thank you for submitting your manuscript to PLOS ONE. After careful consideration, we feel that it has merit but does not fully meet PLOS ONE’s publication criteria as it currently stands. Therefore, we invite you to submit a revised version of the manuscript that addresses the points raised during the review process.

The manuscript has been evaluated by two reviewers, and their comments are available below.

Could you please carefully revise the manuscript to address all comments raised?

We look forward to receiving your revised manuscript.

Kind regards,

Steve Zimmerman, PhD

Senior Editor, PLOS One

Journal Requirements:

Reviewers' comments:

Reviewer's Responses to Questions

**Comments to the Author**

1. Is the manuscript technically sound, and do the data support the conclusions?

Reviewer #1: No

Reviewer #2: Partly

2. Has the statistical analysis been performed appropriately and rigorously? 

Reviewer #1: No

Reviewer #2: Yes

3. Have the authors made all data underlying the findings in their manuscript fully available?

Reviewer #1: Yes

Reviewer #2: Yes

4. Is the manuscript presented in an intelligible fashion and written in standard English?

Reviewer #1: Yes

Reviewer #2: Yes

5. Review Comments to the Author

Reviewer #1: I would like to thank the authors for their work. Their manuscript entitled, “The mediating role of sleep disturbance in the association between perceived neighborhood danger and depressive symptoms in later life”, examines whether sleep disturbances mediate the relationship between perceived neighborhood danger and depressive symptoms among older adults. The authors observed significant relationships between neighborhood danger, sleep disturbance, and mental health; and concluded that sleep disturbance significantly mediated the neighborhood danger and depression relationship Unfortunately, I noted several concerns that I feel significantly hindered the quality of the study, and interpretability of its subsequent findings, which I have outlined in detail below.

Introduction:

1. The authors define “neighborhood danger” as an “individuals’ subjective assessment of how threatening their community and living environment feel”; however, the use of “neighborhood danger” terminology is broad and inconsistent with the literature, which typically focuses on neighborhood safety, neighborhood disorder, neighborhood crime or violence, etc. Moreover, this definition also appears to incorporate “living environment”, which may be interpreted by the reader as the housing experience as well. Is the housing experience relevant? The authors reference the Diez Roux & Mair (2010) article to support this definition; however, I am struggling to locate this neighborhood danger terminology and definition within this article or subsequent articles referenced (citations 1-6). Diez Roux and Mair (2010) provided Figure 1 that is an appropriate representation of components of neighborhood environments and their links with health, of which the authors should consider revising their terminology and incorporate this figure or another conceptual framework/model to describe their independent variable. The National Institutes of Minority Health also has a framework that seem reasonable to include/reference to describe and define constructs (Alvidrez et al., 2019). Please consider these revisions.

Alvidrez, J., Castille, D., Laude-Sharp, M., Rosario, A., & Tabor, D. (2019). The national institute on minority health and health disparities research framework. American journal of public health, 109(S1), S16-S20.

2. In addition to 1 above, sleep disturbance needs to be defined within the introduction. Are the authors focused on objective (e.g., actigraphy measured sleep disturbances) and/or subjective sleep disturbances? Where do sleep disorders fit in within this context? The aspects of objective vs subjective sleep disturbances need lengthier discussion, as well as a clearer rationale for the focus on sleep disturbances and what constitutes this construct.

Method

3. I have numerous questions pertaining to the items selected to represent the predictor and mediator. Specifically, while Cronbach’s alpha reflects marginal (sleep) and high reliability (neighborhood), the rationale for why these items were selected, how these items were selected, and whether these items were from a standardized measure or developed by the NSHAP investigators? These items also do not appear inclusive of the items available in the neighborhood and sleep sections within Wave 3. Specifically, for neighborhood, why were the items selected appropriate to reflect the neighborhood danger construct, and why were items reflecting potential social cohesion excluded? This is particularly true given social cohesion has been linked with perceived neighborhood safety (De Jesus et al., 2011). Variables like social cohesion are not thoughtfully ruled out by the neighborhood danger definition provided within the Introduction.

De Jesus, M., Puleo, E., Shelton, R. C., & Emmons, K. M. (2010). Associations between perceived social environment and neighborhood safety: Health implications. Health & place, 16(5), 1007-1013.

3a. Similarly, within Wave 3, it appears there may be other items that fit into sleep disturbance construct (snoring and sleep apnea). I pose the same questions here regarding sleep disturbance as I did with neighborhood danger.

Given there are numerous questions surrounding two critical constructs within the study, I am concerned about the ability to interpret the study’s findings.

4. Regarding covariates, given sleep disturbances may result from sleeping with one’s partner in the same bed, it is unclear why this was not accounted for as a covariate or some aspect of the analysis. Similarly, years residing in the neighborhood should also be considered as a covariate. Do the authors have rationale for exclusion? If adding these, does it change the findings?

Discussion

5. Within the limitations section, the authors appropriately identified potential multicollinearity across the sleep and depressive symptom items. The sensitivity analysis conducted though does not alleviate concerns regarding multicollinearity that may exist across the mediator and outcome variables. Indeed, there is a large body of evidence denoting strong bi-directional, and comorbid, relationships between sleep and depressive symptoms. I recommend computing variance inflation factors for sleep and depressive symptoms, and presenting those findings with your results.

Reviewer #2: This manuscript examines an important question: the links between neighborhood context, sleep disturbance, and depressive symptoms in later life using a large, nationally representative sample. The study is well motivated, and the analytic approach is generally clear and well organized.

However, several methodological and interpretive issues weaken the conclusions. Concerns about measurement overlap, the marginal reliability of the sleep scale, the reliance on cross-sectional mediation, and the limited description of the sensitivity analyses need more careful attention. In addition, the discussion sometimes goes beyond what the data can reasonably support, especially in its policy and clinical implications.

Major Comments

1. Throughout the abstract and manuscript, the authors describe sleep disturbance as a “pathway” and imply that neighborhood danger undermines mental health. With all variables measured at one wave, this language should be softened substantially. Reverse or reciprocal relationships (e.g., depression influencing sleep or perceptions of safety) are equally plausible and deserve explicit discussion.

2. The manuscript reports the use of the Sobel-Goodman test, with subsequent reference to bootstrap confidence intervals. The analytic description does not clearly specify which method served as the primary inferential approach. Given current standards in mediation analysis, more clarity is needed. The authors should explicitly describe the mediation framework used, specify whether nonparametric bootstrap estimation was the primary method, report the number of bootstrap replications, and clarify the role of the Sobel test in the analysis.

3. “accounting for 27.5% of the total effect”

Although statistically significant, the observed associations, particularly between perceived neighborhood danger and sleep disturbance, are small in magnitude. Emphasizing the proportion of the total effect mediated without sufficient context may overstate the substantive importance of the findings. Effect sizes should be explicitly contextualized, using standardized estimates or interpretive benchmarks. Conclusions regarding practical, clinical, or policy relevance should be scaled to reflect the modest magnitude of observed effects.

4. “the sleep and depressive symptoms scales in the NSHAP data contain overlapping items.”

The overlap between sleep-related items in the sleep disturbance and depressive symptom measures raises an important concern about construct validity. Although a sensitivity analysis is mentioned, the description is brief and does not provide enough detail to evaluate it. The manuscript should clearly specify which items overlap, present the sensitivity analysis results in a table or supplemental material, and directly address how any remaining measurement overlap may have influenced the findings.

5. “the mediating role of sleep disturbance remained consistent across all demographic subgroups”

This statement is not supported by results presented in the manuscript. No subgroup-specific estimates or formal interaction tests are shown. The authors should either present formal subgroup or interaction analyses to support this claim or revise the text to avoid statements about consistency across demographic subgroups.

6. “α = .68”

The internal consistency of the sleep disturbance scale is marginal. Given that sleep disturbance is central to the analytic model, this limitation is important. The implications of limited scale reliability should be discussed more explicitly, and conclusions regarding the role of sleep disturbance should be interpreted with appropriate caution.

7. “suggesting that sleep disturbance may serve as an explanatory pathway”

Coefficient attenuation following covariate adjustment is consistent with an indirect association but does not, on its own, establish mediation. Interpretation should be revised to emphasize statistical consistency with an indirect association model rather than explanatory or causal inference.

8. “R² .14***”

R² values do not have associated p-values, and the inclusion of significance indicators is misleading. Please clarify what this asterisk means?

9. “advances our understanding of the mechanisms”

The study does not empirically test biological or behavioral mechanisms. Mechanistic interpretations should be framed as conceptual or theory-informed rather than empirically demonstrated.

10. “Routine sleep screening … timely prevention of depression”

The manuscript discusses intervention and prevention implications without directly evaluating screening or treatment outcomes. Policy and intervention implications should be reframed as potential directions for future research rather than evidence-based recommendations.

11. “sleep disturbance” vs. “sleep problems”

Inconsistent terminology reduces clarity. Terminology should be standardized throughout the manuscript.

12. Reference not cited:

Neighborhood support as a protective factor for cognition: Associations with sleep, depression, and stress (Singh RK et al. https://doi.org/10.1002/alz.70940)

The manuscript examines perceived neighborhood danger in relation to sleep disturbance and depressive symptoms, yet it does not engage with recent work demonstrating that positive neighborhood characteristics, such as neighborhood support, are associated with cognition through related psychosocial and sleep-related pathways. In particular, the study by Singh et al. directly examines neighborhood context alongside sleep, depression, and stress, thereby providing a closely aligned conceptual and empirical framework. Omitting this work limits the completeness of the literature review and presents neighborhood effects primarily in deficit-oriented terms, without acknowledging evidence that supportive neighborhood environments may confer protective effects through overlapping mechanisms.

The authors should consider citing and briefly discussing this study to situate their findings within the broader literature on neighborhood context, sleep, and mental health. It will strengthen the theoretical framing and clarify how perceived neighborhood danger relates to both adverse and protective neighborhood processes described in prior research.

12. “Findings highlight the importance of incorporating sleep-focused interventions into mental health prevention strategies…”

“Routine sleep screening … early identification and timely prevention of depression…”

“Healthcare systems and insurance programs may consider covering evidence-based sleep interventions…”

The discussion draws broad clinical and policy implications that are not directly examined in the analysis. The study evaluates associations among perceived neighborhood danger, sleep disturbance, and depressive symptoms measured at a single time point. It does not assess whether sleep disturbance predicts future depression, functions as a practical screening marker, or represents an effective point of intervention. As a result, statements regarding prevention strategies, screening practices, and healthcare coverage are insufficiently grounded in the presented data and may be interpreted as extending beyond the scope of the findings. The implications section would benefit from tighter alignment with the study’s analytic focus. Statements related to screening, prevention, and intervention should be reframed as potential areas for future investigation rather than inferences drawn from the current results.

6. PLOS authors have the option to publish the peer review history of their article (what does this mean?). If published, this will include your full peer review and any attached files.

Reviewer #1: No

Reviewer #2: No

---

## [Author Response · Author response to Decision Letter 1]

10 Mar 2026

March 10, 2026

Dr. Steve Zimmerman

Editor, PLOS One

em@editorialmanager.com

Dear Dr. Zimmerman,

Enclosed is our revision of the manuscript entitled “The Mediating Role of Sleep Disturbance in the Association between Perceived Neighborhood Danger and Depressive Symptoms in Later Life.” We sincerely appreciate the thoughtful comments on our work from the reviewers. In this letter, we respond to each of the concerns raised. Revised passages in the manuscript are noted in red fonts.

Comment from the Reviewer 1

1. The authors define “neighborhood danger” as an “individuals’ subjective assessment of how threatening their community and living environment feel”; however, the use of “neighborhood danger” terminology is broad and inconsistent with the literature, which typically focuses on neighborhood safety, neighborhood disorder, neighborhood crime or violence, etc. Moreover, this definition also appears to incorporate “living environment”, which may be interpreted by the reader as the housing experience as well. Is the housing experience relevant? The authors reference the Diez Roux & Mair (2010) article to support this definition; however, I am struggling to locate this neighborhood danger terminology and definition within this article or subsequent articles referenced (citations 1-6). Diez Roux and Mair (2010) provided Figure 1 that is an appropriate representation of components of neighborhood environments and their links with health, of which the authors should consider revising their terminology and incorporate this figure or another conceptual framework/model to describe their independent variable. The National Institutes of Minority Health also has a framework that seem reasonable to include/reference to describe and define constructs (Alvidrez et al., 2019). Please consider these revisions.

Response: We agree that our independent variable should be more clearly defined and situated within the existing literature. In response to the Reviewer’s suggestions regarding theoretical frameworks and conceptualization, we have revised the text as follows.

“Research on health determinants has increasingly recognized the importance of factors that extend beyond the individual level. The NIMHD Research Framework [1] calls for greater attention to community-level determinants of health, and the model proposed by Diez Roux and Mair [2] illustrates how neighborhood social environments influence health through behavioral mechanisms and stress-related processes. In line with these frameworks, the present study focuses on perceived neighborhood danger as a key neighborhood environmental factor and addresses its health impact in later life.

Defined as residents’ subjective perceptions of danger in their local environment [3], perceived neighborhood danger is particularly relevant to sleep and mental health because it reflects individuals’ lived experiences and cognitive appraisals of threat. When individuals perceive their surroundings as unsafe or dangerous, they often experience heightened vigilance and anxiety, which can undermine sleep quality and overall well-being. It is noteworthy that subjective perceptions of the neighborhood environment exert stronger influences on mental health than objective indicators such as crime statistics (e.g., crime and arrest rates) and physical disorder (e.g., graffiti, vandalism, abandoned buildings) [4-6]. Furthermore, older adults are known to be more susceptible to neighborhood influences than their younger counterparts, as they spend more time within their neighborhoods [3]. However, the pathways linking perceived neighborhood danger to depressive symptoms in later life remain largely unexplored.”

2. In addition to 1 above, sleep disturbance needs to be defined within the introduction. Are the authors focused on objective (e.g., actigraphy measured sleep disturbances) and/or subjective sleep disturbances? Where do sleep disorders fit in within this context? The aspects of objective vs subjective sleep disturbances need lengthier discussion, as well as a clearer rationale for the focus on sleep disturbances and what constitutes this construct.

Response: We agree that the sleep construct required clearer definition. After revisiting the sleep literature, particularly studies using the NSHAP data, we decided to use the term "insomnia symptoms" instead of “sleep disturbance.” The three items included in our study (difficulty falling asleep, trouble with waking up during the night, and trouble with waking up too early and not being able to fall asleep again) correspond to the cardinal symptoms of insomnia, defined by DSM-5. We also note that this terminology has been used in prior studies employing the same NSHAP items (e.g., Chen et al., 2015). In the revised introduction, we establish from the outset why insomnia symptoms serve as the focal sleep construct in this study, situating it within the stress and neighborhood health literature before presenting it as a potential mediating factor. In the revised manuscript, we have clarified the sleep construct and its measurement, providing a clear rationale and appropriate citations. Additionally, we acknowledge the importance of incorporating objective sleep measures (e.g., actigraphy) and have noted this as a limitation of the current study. The revised sections are provided below.

“Filling this gap, we conceptualize sleep problems as an intermediate factor in the association between perceived neighborhood danger and depressive symptoms. Among various types of sleep problems, we focus on insomnia symptoms given their high prevalence and relevance to the current investigation. Referring to difficulties initiating or maintaining sleep, or experiencing poor-quality sleep [7], insomnia symptoms are frequently experienced by individuals who undergo age-related changes in sleep and circadian rhythms [8]. The particular link between insomnia symptoms and depression has been widely documented across diverse groups of older adults [9, 10]. Neighborhood research also suggests that worries and concerns related to neighborhood danger make older individuals prone to insomnia symptoms [11, 12]. Taken together, it is plausible to anticipate that the level of neighborhood danger perceived by older individuals is associated with insomnia symptoms, which in turn lead to depressive symptoms. This mediating conceptualization aligns with stress literature and pathway models linking neighborhood environments to mental health outcomes [13-17]. Findings from this investigation would help improve understanding of how neighborhood environments shape older adults’ sleep and mental health, as well as the psychological mechanisms underlying these relationships, and may offer implications for health promotion strategies.”

"Insomnia symptoms were assessed using three items ("trouble falling asleep," "trouble waking up during the night," and "trouble waking up too early and not being able to fall asleep again"), corresponding to the cardinal clinical features of insomnia defined in the Diagnostic and Statistical Manual of Mental Disorders, Fifth Edition (DSM-5) [22]. These items have been used to assess insomnia symptoms in prior NSHAP research [23]. Respondents reported the frequency of each symptom on a 4-point scale (0 = never to 3 = most of the time). Total scores ranged from 0 to 9, with higher scores indicating more frequent insomnia symptoms. The scale demonstrated marginally acceptable internal consistency (α = .68).”

"Third, insomnia symptoms were assessed via self-report without objective measurement. Future research should incorporate objective measures, such as actigraphy, to complement self-report data."

3. Methods: I have numerous questions pertaining to the items selected to represent the predictor and mediator. Specifically, while Cronbach’s alpha reflects marginal (sleep) and high reliability (neighborhood), the rationale for why these items were selected, how these items were selected, and whether these items were from a standardized measure or developed by the NSHAP investigators? These items also do not appear inclusive of the items available in the neighborhood and sleep sections within Wave 3. Specifically, for neighborhood, why were the items selected appropriate to reflect the neighborhood danger construct, and why were items reflecting potential social cohesion excluded? This is particularly true given social cohesion has been linked with perceived neighborhood safety (De Jesus et al., 2011). Variables like social cohesion are not thoughtfully ruled out by the neighborhood danger definition provided within the Introduction.

Response: As noted in our response above, we have redefined our primary study constructs and provided a clear rationale for focusing on perceived neighborhood danger and insomnia symptoms among the broader dimensions of neighborhood characteristics and sleep. Although perceived danger is conceptually related to other neighborhood features, including social cohesion, we chose to conduct a more focused assessment to maintain conceptual clarity and analytic precision.

With Cronbach’s alpha values ranging from marginally acceptable to highly satisfactory, both measures demonstrated adequate internal consistency for the proposed research model. We have also cited prior studies that employed the same measures to further support their validity and use in this context. Specifically, we have added the following description in the Methods section to clarify the basis for the insomnia symptom items:

“Insomnia symptoms were assessed … corresponding to the cardinal clinical features of insomnia defined in the Diagnostic and Statistical Manual of Mental Disorders, Fifth Edition (DSM-5) [22]. These items have been used to assess insomnia symptoms in prior NSHAP research [23].”

3a. Methods: Similarly, within Wave 3, it appears there may be other items that fit into sleep disturbance construct (snoring and sleep apnea). I pose the same questions here regarding sleep disturbance as I did with neighborhood danger. Given there are numerous questions surrounding two critical constructs within the study, I am concerned about the ability to interpret the study’s findings.

Response: As noted in our response above comment #2, we adopted a targeted approach rather than a broad, general assessment. Specifically, we focused on perceived neighborhood danger and insomnia symptoms and provided a clear rationale for selecting those constructs. We would like to note that snoring and sleep apnea are primarily physiological conditions with distinct etiologies and may therefore be less directly related to our study’s focus on subjective sleep experiences that are sensitive to environmental and psychological stressors.

4. Covariates: Regarding covariates, given sleep disturbances may result from sleeping with one’s partner in the same bed, it is unclear why this was not accounted for as a covariate or some aspect of the analysis. Similarly, years residing in the neighborhood should also be considered as a covariate. Do the authors have rationale for exclusion? If adding these, does it change the findings?

Response: We thank the Reviewer for this suggestion. Both bed-sharing partner and residential duration variables were available in the dataset. Having a bed-sharing partner was highly correlated with being married (correlation coefficient = .65). Due to concerns about collinearity, we retained only marital status, as it serves as a broader sociodemographic indicator.

There were no collinearity concerns regarding residential duration, and the regression models were re-run including this variable. The revised manuscript presents the results of these updated models. We would like to note that the main findings remain unchanged.

5. Discussion: Within the limitations section, the authors appropriately identified potential multicollinearity across the sleep and depressive symptom items. The sensitivity analysis conducted though does not alleviate concerns regarding multicollinearity that may exist across the mediator and outcome variables. Indeed, there is a large body of evidence denoting strong bi-directional, and comorbid, relationships between sleep and depressive symptoms. I recommend computing variance inflation factors for sleep and depressive symptoms and presenting those findings with your results.

Response: Our revised manuscript now includes information on VIF. We have also noted the potential of bi-directional, and comorbid, relationship between sleep and mental health in the limitation section.

“Table 3 presents the regression models of depressive symptoms. Variance inflation factors (VIF) for all predictors ranged from 1.03 to 1.22, indicating no multicollinearity concerns.”

“Several limitations should be considered when interpreting these findings. First, the cross-sectional design limits causal inference, and reverse or reciprocal relationships are possible; longitudinal studies are needed to clarify directionality.”

Comment from the Reviewer 2

1. Throughout: Throughout the abstract and manuscript, the authors describe sleep disturbance as a “pathway” and imply that neighborhood danger undermines mental health. With all variables measured at one wave, this language should be softened substantially. Reverse or reciprocal relationships (e.g., depression influencing sleep or perceptions of safety) are equally plausible and deserve explicit discussion.

Response: We have carefully revised the manuscript to soften causal language throughout. As the Discussion section underwent substantial revisions in tone and framing, we provide the revised opening below:

“The environments in which we live influence our health and quality of life in ways that extend well beyond what is outwardly visible. For older adults, the surrounding environment can foster a sense of security and belonging or, conversely, pose a risk to mental health [3]. In neighborhoods perceived as dangerous, sleep problems such as insomnia symptoms may arise as a response to heightened vigilance and stress, which can increase susceptibility to depressive symptoms. Guided by theoretical frameworks from environmental health and the stress process model [1, 2, 13-17], we hypothesized that the mental health impact of perceived neighborhood danger would be mediated by increased insomnia symptoms. Our findings provide supportive evidence for this pathway.”

2. Methods: The manuscript reports the use of the Sobel-Goodman test, with subsequent reference to bootstrap confidence intervals. The analytic description does not clearly specify which method served as the primary inferential approach. Given current standards in mediation analysis, more clarity is needed. The authors should explicitly describe the mediation framework used, specify whether nonparametric bootstrap estimation was the primary method, report the number of bootstrap replications, and clarify the role of the Sobel test in the analysis.

Response: We sincerely appreciate this important methodological recommendation. To improve clarity, we re-conducted the mediation analysis using the PROCESS macro for SPSS with bootstrap estimation (5,000 resamples). The Analytic Strategy section has been revised as follows.

“Ultimately, the indirect effect model was tested using PROCESS macro [27]. The 95% confidence interval was estimated using 5,000 bootstrap resamples. Analyses were performed using SPSS Statistics 29 (IBM Corp., Armonk, NY).”

3. Results/Discussion: “accounting for 27.5% of the total effect” Although statistically significant, the observed associations, particularly between perceived neighborhood danger and sleep disturbance, are small in magnitude. Emphasizing the proportion of the total effect mediated without sufficient context may overstate the substantive importance of the findings. Effect sizes should be explicitly contextualized, using standardized estimates or interpretive benchmarks. Conclusions regarding practical, clinical, or policy relevance should be scaled to reflect the modest magnitude of observed effects.

Response: We thank the Reviewer for this thoughtful comment regarding the magnitude and interpretation of the observed effects. We agree that, although statistically

---

## [Decision Letter · Decision Letter 1]

20 Apr 2026

The mediating role of insomnia symptoms in the association between perceived neighborhood danger and depressive symptoms in later life

PONE-D-25-66548R1

Dear Dr. Jang,

We’re pleased to inform you that your manuscript has been judged scientifically suitable for publication and will be formally accepted for publication once it meets all outstanding technical requirements.

Kind regards,

Annesha Sil, Ph.D.

Staff Editor

PLOS One

Additional Editor Comments (optional):

Reviewers' comments:

Reviewer's Responses to Questions

**Comments to the Author**

1. If the authors have adequately addressed your comments raised in a previous round of review and you feel that this manuscript is now acceptable for publication, you may indicate that here to bypass the “Comments to the Author” section, enter your conflict of interest statement in the “Confidential to Editor” section, and submit your "Accept" recommendation.

Reviewer #1: All comments have been addressed

Reviewer #2: All comments have been addressed

2. Is the manuscript technically sound, and do the data support the conclusions?

Reviewer #1: Yes

Reviewer #2: Yes

3. Has the statistical analysis been performed appropriately and rigorously? 

Reviewer #1: Yes

Reviewer #2: Yes

4. Have the authors made all data underlying the findings in their manuscript fully available?

Reviewer #1: Yes

Reviewer #2: Yes

5. Is the manuscript presented in an intelligible fashion and written in standard English?

Reviewer #1: Yes

Reviewer #2: Yes

6. Review Comments to the Author

Reviewer #1: I appreciate the author(s) time to consider and address the feedback provided for the first draft. This version is markedly clearer. I have no further comments and/or requests for edits at this time.

Reviewer #2: The authors have satisfactorily addressed all the comments and concerns. I have no further comments.

7. PLOS authors have the option to publish the peer review history of their article (what does this mean?). If published, this will include your full peer review and any attached files.

Reviewer #1: No

Reviewer #2: No

---

## [Editor Report · Acceptance letter]

PONE-D-25-66548R1

PLOS One

Dear Dr. Jang,

I'm pleased to inform you that your manuscript has been deemed suitable for publication in PLOS One. Congratulations! Your manuscript is now being handed over to our production team.

Kind regards,

on behalf of

Dr Annesha Sil

Staff Editor

PLOS One